# AV-NeRF: Learning Neural Fields for Real-World Audio-Visual Scene Synthesis

**Susan Liang**[1]    **Chao Huang**[1]    **Yapeng Tian**[1]
**Anurag Kumar**[2]    **Chenliang Xu**[1]
[1]University of Rochester    [2]Meta Reality Labs Research

## Abstract

Can machines recording an audio-visual scene produce realistic, matching audio-visual experiences at novel positions and novel view directions? We answer it by studying a new task—real-world audio-visual scene synthesis—and a first-of-its-kind NeRF-based approach for multimodal learning. Concretely, given a video recording of an audio-visual scene, the task is to synthesize new videos with spatial audios along arbitrary novel camera trajectories in that scene. We propose an acoustic-aware audio generation module that integrates prior knowledge of audio propagation into NeRF, in which we implicitly associate audio generation with the 3D geometry and material properties of a visual environment. Furthermore, we present a coordinate transformation module that expresses a view direction relative to the sound source, enabling the model to learn sound source-centric acoustic fields. To facilitate the study of this new task, we collect a high-quality Real-World Audio-Visual Scene (RWAVS) dataset. We demonstrate the advantages of our method on this real-world dataset and the simulation-based SoundSpaces dataset. We recommend that readers visit our project page for convincing comparisons: `https://liangsusan-git.github.io/project/avnerf/`.

## 1   Introduction

We study a new task, *real-world audio-visual scene synthesis*, to generate target videos and audios along novel camera trajectories from source audio-visual recordings of known trajectories. By learning from real-world source videos with binaural audio, we aim to generate target video frames and spatial audios that exhibit consistency with the given camera trajectory visually and acoustically. This consistency ensures perceptual realism and immersion, enriching the overall user experience.

As far as we know, attempts in the audio-visual learning literature [1–11] have yet to succeed in solving this challenging task thus far. Although there are similar works [12–15], these methods have constraints that limit their ability to solve this new task. Luo et al. [12] propose neural acoustic fields to model sound propagation in a room. Su et al. [13] introduce representing audio scenes by disentangling the scene's geometry features. These methods are tailored for estimating room impulse response signals in a simulation environment that are difficult to obtain in a real-world scene. Concurrent to our work, ViGAS proposed by Chen et al. [15] learns to synthesize new sounds by inferring the audio-visual cues. However, ViGAS is limited to a few viewpoints for audio generation.

We introduce AV-NeRF, a novel NeRF-based method of synthesizing real-world audio-visual scenes. AV-NeRF enables the generation of videos and spatial audios, following arbitrary camera trajectories. It utilizes source videos and camera poses as references. AV-NeRF consists of two branches: A-NeRF, which learns the acoustic fields of an environment, and V-NeRF, which models color and density fields. We represent a static audio field as a continuous function using A-NeRF, which takes the listener's position and head direction as input. A-NeRF effectively models the energy decay of sound as the sound travels from the source to the listener by correlating the listener's position with the

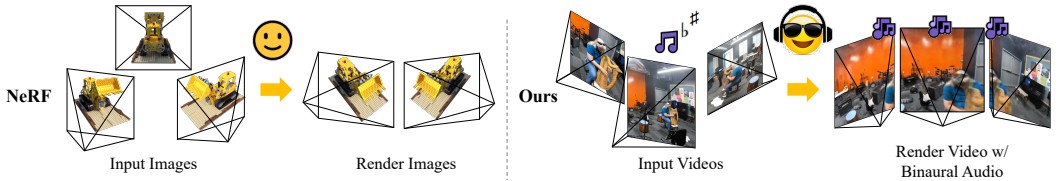

Figure 1: NeRF learns to render visual scenes from novel viewpoints. Beyond visual rendering, we introduce AV-NeRF, a method for synthesizing audio-visual scenes with video frames and binaural audios along new camera trajectories. Integrating coherent sight and sound creates an immersive and realistic perceptual experience for users.

energy reduction effect. Moreover, A-NeRF accounts for the impact of the listener's orientation on sound perception by establishing a correlation between the audio channel difference and the listener's direction. Consequently, A-NeRF facilitates **distance-sensitive** and **direction-aware** audio synthesis, resulting in the generation of realistic binaural sound. We utilize the vanilla NeRF [16] as our V-NeRF for vision rendering since the rendering of the purely visual world has been extensively researched [17, 16, 18–20] and is not the primary focus of our work.

To further enhance synthesis quality, we introduce an acoustic-aware audio generation method and a coordinate transformation mechanism. Considering that the 3D geometry and material properties of an environment determine sound propagation [21, 22, 4, 23], we propose an acoustic-aware audio generation module called AV-Mapper. AV-Mapper extracts geometry and material information from V-NeRF and feed them to A-NeRF, enabling A-NeRF to generate audios with acoustic awareness. In visual space, NeRF [16] expresses the viewing direction $(\theta, \phi)$ using an absolute coordinate system, where the directions of two parallel light rays are represented identically. However, in auditory space, this expression method is unsuitable as it ignores the spatial relationship between the sound emitter and receiver. Human perception of sound source relies on the relative direction of the sound source to the head orientation. Given this observation, we propose a coordinate transform mechanism that expresses the viewing direction relative to the sound source. This approach encourages our model to learn sound source-centric acoustic fields.

Our work represents the initial step towards addressing the audio-visual scene synthesis task in real-world settings. As such, there is currently no publicly available dataset that fulfills our specific requirements. To facilitate our study, we curated a high-quality audio-visual dataset called RWAVS (Real-World Audio-Visual Synthesis), which encompasses multimodal data, including camera poses, video frames, and realistic binaural (stereo) audios. In order to enhance the diversity of our dataset, we collected data from various environments, ranging from offices and apartments to houses and even outdoor spaces. We will release this dataset to the research community. Experiments on the RWAVS and the synthetic SoundSpaces datasets can validate the effectiveness of our proposed approach. It demonstrates AV-NeRF's capability of synthesizing novel-view audio-visual scenes in a range of environments, including both real-world and synthetic, as well as indoor and outdoor settings.

In summary, our contributions include: (1) proposing a novel approach to synthesize audio-visual scenes in real-world environments at novel camera poses; (2) introducing a new acoustic-aware audio generation method that incorporates our prior knowledge of sound propagation; (3) suggesting a coordinate transformation mechanism to effectively express sound direction; (4) curating a high-quality audio-visual dataset for benchmarking this task; and (5) demonstrating the advantages of our method through quantitative and qualitative evaluations.

## 2   Related Work

Our work is closely related to several areas, including novel-view synthesis, vision-guided audio spatialization, simulation-based acoustic synthesis, and learning-based audio generation. We elaborate on each of these topics in the following section.

**Neural Fields.** Our method is based on neural fields, particularly Neural Radiance Fields (NeRF) [16]. NeRF employs MLPs to learn an implicit and continuous representation of visual scenes, enabling the synthesis of novel views. Building upon NeRF's foundation, several works have extended its applicability to broader domains, including in-the-wild images [24], (silent) video [25–27], audio

[12, 13], and audio-visual [14] content. Luo et al. [12] propose neural acoustic fields (NAF) to capture the sound propagation in an environment, and Su et al. [13] introduce disentangling a scene's geometry features for audio scene representation (INRAS). Despite the compelling results achieved by NAF and INRAS, their application in the real world is hindered by their reliance on ground-truth impulse response signals that are simulated in synthetic environments. Furthermore, the discretization of position and direction in these approaches restricts the range of camera poses they can handle, whereas our model can handle continuous position and direction queries. Du et al. [14] proposed a manifold learning method that maps vectors from a latent space to audio and image spaces. Although the learned manifold allows for audio-visual interpolation, the model lacks support for controllable audio-visual generation. In contrast, our method focuses on learning audio-visual representations that are explicitly conditioned on spatial coordinates, enabling controllable generation.

**Visually Informed Spatial Audio Generation.** Considering that images often reveal the position of a sound source and/or the structure of a scene, many visually informed audio spatialization approaches [7, 28–31, 15] have been proposed. Among them, Gao and Grauman [7] focus on normal field-of-view videos and binaural audios. Zhou et al. [28] propose a unified framework to solve the sound separation and stereo sound generation at the same time. Xu et al. [29] propose a Pseudo2Binaural pipeline that augments audio data using HRIR function [32]. Most recently, Chen et al. [15] proposed the Visually-Guided Acoustic Synthesis (ViGAS) method for generating novel-view audios. However, AV-NeRF differs from this method in two ways: (1) ViGAS relies on ground-truth images for audio transformation, whereas AV-NeRF addresses the absence of ground-truth images by utilizing rendered images from learned vision fields; (2) ViGAS only supports a limited number of viewpoints for audio synthesis, while AV-NeRF has the capability to render novel videos at arbitrary camera poses.

**Geometry and Material Based Acoustic Simulation.** Several works [4, 33, 22, 34, 23] focus on simulating acoustic environments by modeling their 3D geometry and material properties. SoundSpaces, proposed by Chen et al. [4], is an audio platform that calculates the room impulse response signals for discrete listener and sound source positions. Extending this work, Chen et al. [33] introduce SoundSpaces 2.0, which enables acoustic generation for arbitrary microphone locations. Tang et al. [23] propose calibrating geometric acoustic ray-tracing with a finite-difference time-domain wave solver to compute high-quality impulse responses. Additionally, Li et al. [22] propose a method that blends the early reverberation portion, modeled using geometric acoustic simulation and frequency modulation, with a late reverberation tail extracted from recorded impulse responses. In contrast to these simulation methods, our approach implicitly leverages learned geometry and material information for audio-visual scene synthesis.

**Learning Based Audio Generation.** Learning-based methods exploit the powerful modeling ability of neural networks to synthesize audios. Tang et al. [34] propose the use of neural networks to estimate reverberation time ($T_{60}$) and equalization (EQ). Ratnarajah et al. [35, 36, 37] employ generative adversarial networks (GANs) to supervise realistic room impulse response generation. Richard et al. [38] introduce a binauralization network that models the sound propagation linearly and nonlinearly. Recent research has also investigated learning from noisy data [39] and few-shot data [40].

## 3  Task Definition

The real-world audio-visual scene synthesis task is to generate visual frames and corresponding binaural audios for arbitrary camera trajectories. Given a static environment $E$ and multiple observations $O = \{O_1, O_2, \ldots, O_N\}$ of this environment, where $N$ is the number of training samples and each observation $O_i$ includes the camera pose $p = (x, y, z, \theta, \phi)$, the mono source audio clip $a_s$, the recorded binaural audio $a_t$, and the image $I$, this task aims to synthesize a new binaural audio $a_t^*$ and a novel view $I^*$ based on a camera pose query $p^*$ and a source audio $a_s^*$, where $p^*$ is distinct from the camera poses in all observations. This process can be formatted as:

$$(a_t^*, I^*) = f(p^*, a_s^* | O, E) \ , \tag{1}$$

where $f$ is a mapping function. Combining predicted $a_t^*$ and $I^*$ of all poses in the queried camera trajectory, we can synthesize a realistic video with spatial audio. This task poses several challenges: (1) $p^*$ is a novel viewpoint distinct from all observed viewpoints, (2) the synthesized binaural audio $a_t^*$ is expected to exhibit rich spatial effects contributing to its realism, (3) during inference, no observation $O$ is accessible and the mapping function $f$ relies solely on the learned fields for scene synthesis. It should be noted that the position of the sound source is known in the environment $E$.

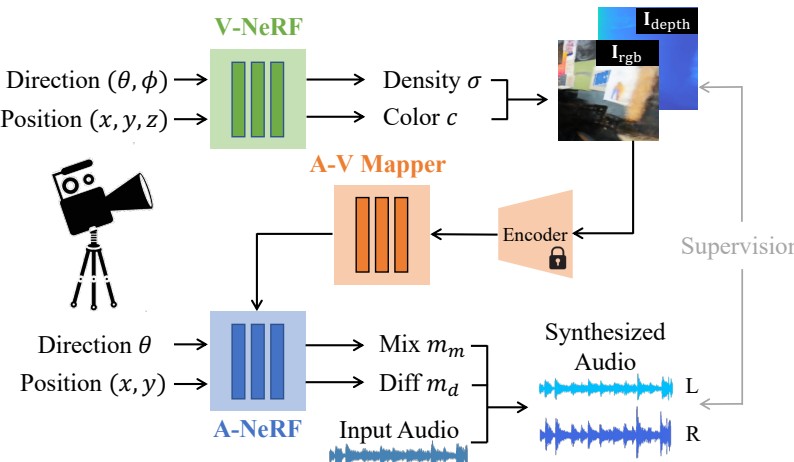

Figure 2: The pipeline of our method. Given the position $(x, y, z)$ and viewing direction $(\theta, \phi)$ of a listener, our method can render an image the listener would see and the corresponding binaural audio the listener would hear. Our model consists of V-NeRF, A-NeRF, and AV-Mapper. A-NeRF learns to generate acoustic masks, V-NeRF learns to generate visual frames, and AV-Mapper is optimized to integrate geometry and material information extracted from V-NeRF into A-NeRF.

## 4 Method

Our method aims to learn neural fields that can synthesize real-world audio-visual scenes from novel poses. The entire pipeline is illustrated in Figure 2. Our model comprises three trainable modules: V-NeRF, A-NeRF, and AV-Mapper. A-NeRF is responsible for generating acoustic masks, V-NeRF focuses on generating visual frames, and AV-Mapper is optimized to extract geometry and material information from V-NeRF, integrating this information into A-NeRF.

### 4.1 V-NeRF

NeRF [16] uses a Multi-Layer Perceptron (MLP) to implicitly and continuously represent a visual scene. It effectively learns a mapping from camera poses to colors and densities:

$$\text{NeRF} : (x, y, z, \theta, \phi) \rightarrow (\mathbf{c}, \sigma) \ , \tag{2}$$

where $\mathbf{X} = (x, y, z)$ is the 3D position, $\mathbf{d} = (\theta, \phi)$ is the direction, $\mathbf{c} = (r, g, b)$ is the color, and $\sigma$ is the density. In order to achieve view-dependent color rendering $\mathbf{c}$ and ensure multi-view consistency, [16] introduces two MLP components within NeRF. The first MLP maps a 3D coordinate $(x, y, z)$ to density $\sigma$ and a corresponding feature vector. The second MLP then takes the feature vector and 2D direction $(\theta, \phi)$ as inputs, producing the color $\mathbf{c}$. This process is illustrated in Fig. 3. NeRF then uses the volume rendering method [41] to generate the color of any ray $\mathbf{r}(t) = \mathbf{o} + t\mathbf{d}$ as it traverses through the visual scene, bounded by near $t_n$ and far $t_f$:

$$C(\mathbf{r}) = \int_{t_n}^{t_f} T(t)\sigma(\mathbf{r}(t))\mathbf{c}(\mathbf{r}(t), \mathbf{d})dt \ , \tag{3}$$

where $T(t) = \exp(-\int_{t_n}^{t} \sigma(\mathbf{r}(s))ds)$. The depth of any ray $\mathbf{r}$ can be calculated similarly by replacing $\mathbf{c}(\mathbf{r}(t), \mathbf{d})$ with $t$:

$$D(\mathbf{r}) = \int_{t_n}^{t_f} T(t)\sigma(\mathbf{r}(t))tdt \ . \tag{4}$$

These continuous integrals are estimated by quadrature in practice [42]. Subsequently, RGB and depth images can be rendered for any given camera pose using V-NeRF (Fig. 3). We apply positional encoding to all input coordinates to preserve the high-frequency information of the generated images.

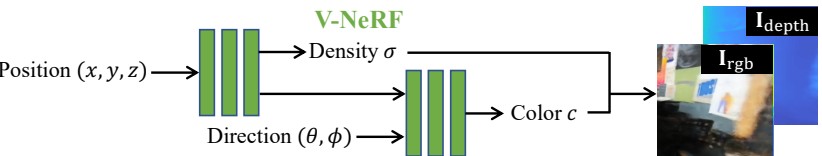

Figure 3: The pipeline of V-NeRF. Given the 3D position $(x, y, z)$ and direction $(\theta, \phi)$ of a camera pose, V-NeRF can render corresponding RGB and depth images. V-NeRF consists of two MLPs with the first MLP estimating density $\sigma$ and the second one predicting color $\mathbf{c}$.

## 4.2 A-NeRF

The goal of A-NeRF is to learn a neural acoustic representation capable of mapping 5D coordinates $(x, y, z, \theta, \phi)$ to corresponding acoustic masks $\mathbf{m}_m, \mathbf{m}_d \in \mathcal{R}^F$, where $\mathbf{m}_m$ quantifies the change in audio magnitude with respect to the position $(x, y, z)$ while $\mathbf{m}_d$ characterizes the impact of the direction $(\theta, \phi)$ on the channel difference of binaural audios, and $F$ is the number of frequency bins:

$$\text{NeRF} : (x, y, z, \theta, \phi) \rightarrow (\mathbf{m}_m, \mathbf{m}_d) \ . \tag{5}$$

After training A-NeRF, we can synthesize binaural sound that contains rich spatial and acoustic information by fusing the magnitude of any input audio with predicted $\mathbf{m}_m$ and $\mathbf{m}_d$.

In practice, we simplify the problem of audio synthesis by discarding the $z$ axis and the $\phi$ direction, which represents coordinates related to height, and instead focus on the 2D space. On RWAVS and SoundSpaces datasets, we observe that the sound received by the listener exhibits more distinct variations along the horizontal plane, as opposed to the vertical axis. Fig. 4 depicts the pipeline of audio synthesis. Similar to V-NeRF, A-NeRF consists of two MLP components that parameterize different aspects of acoustic fields. The first MLP takes as input the 2D position $(x, y)$ and frequency query $f \in [0, F]$, producing a mixture mask $\mathbf{m}_m$ for frequency $f$ and a feature vector. The mask $\mathbf{m}_m$ quantifies the change in audio magnitude based on the distance between the sound source and the sound receiver. The feature vector serves as an implicit embedding of the input information and the acoustic scene. Then we concatenate the feature vector with the transformed direction $\theta'$ (explained in Sec. 4.4) and pass them to the second MLP. The second MLP associates the direction $\theta'$ with the channel difference of binaural audios. Given a fixed sound emitter, the difference between two channels of the received audio changes as the direction $\theta'$ of the sound receiver varies. For instance, when the receiver faces the emitter, the energy of the two channels will be approximately equal. Conversely, when the emitter is on the left side of the receiver, the left channel will exhibit higher energy compared to the right channel. The second MLP is optimized to generate a difference mask $\mathbf{m}_d$ that characterizes such direction influence. We query A-NeRF with all frequencies $f$ within $[0, F]$ to generate the complete masks $\mathbf{m}_m$ and $\mathbf{m}_d$.

After obtaining the mixture mask $\mathbf{m}_m$ and the difference mask $\mathbf{m}_d$, we can synthesize binaural audios by composing these masks and input audios (Fig. 4). For an input source audio $a_s$, we initially employ the short-time Fourier transform (STFT) to calculate the magnitude $s_s \in \mathcal{R}^{F \times W}$, where $W$ represents the number of time frames and $F$ denotes the number of frequency bins. We then multiply $s_s$ with mask $\mathbf{m}_m$ to obtain the mixed magnitude $s_m$. Additionally, we multiply $s_m$ and mask $\mathbf{m}_d$ to predict the difference magnitude $s_d$. Then we add mixture magnitude $s_m$ and difference magnitude $s_d$ to compute the magnitude of left channel $s_l$, and subtract $s_d$ from $s_m$ to compute the magnitude of right channel $s_r$. We further refine $s_l$ and $s_r$ using 2D convolution. Finally, we apply Inverse STFT to the predicted $s_l$ and $s_r$, respectively, to synthesize the binaural audio $a_t$. Because A-NeRF operates on the magnitude, we use the phase information of the source audio for the Inverse STFT process. Instead of estimating two channels of the target audio directly, we predict the mixture and the difference between the two channels of the target audio. Gao and Grauman [7] suggest that direct two-channel predictions can lead to shortcut learning in the spatialization network.

## 4.3 AV-Mapper

Given the fact that 3D geometry and material property determine sound propagation in an environment (*prior knowledge*), we propose an acoustic-aware audio generation method that integrates color and depth information estimated by V-NeRF with A-NeRF. When V-NeRF learns to represent visual scenes, it can learn the color $\mathbf{c}$ and density $\sigma$ function of the environment, thanks to the multi-view

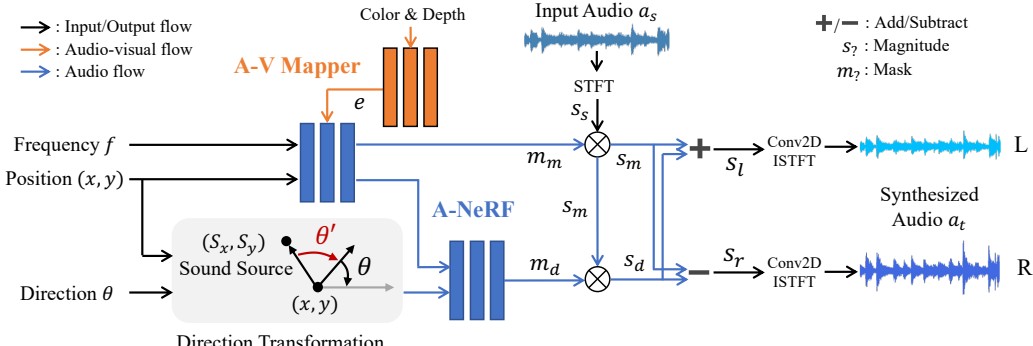

Figure 4: The pipeline of A-NeRF. Given the 2D position $(x, y)$ and direction $\theta$ of a camera pose, A-NeRF can render mixture mask $\mathbf{m}_m$, and difference mask $\mathbf{m}_d$, which then be used for audio synthesis. A-NeRF is also composed of two MLPs, with the first predicting mixture mask $\mathbf{m}_m$ and the second estimating difference mask $\mathbf{m}_d$. The direction $\theta$ is transformed relatively to the sound source prior to inputting into A-NeRF.

consistency constraint. Utilizing volume rendering [16, 41], we can synthesize the RGB and depth image for any given camera pose. The RGB image captures the semantics and category of each object, implicitly indicating their material properties. And the depth image depicts the geometric structure of the environment. By providing A-NeRF with a combination of RGB and depth images, we can offer rich environmental knowledge to A-NeRF.

Specifically, when synthesizing the binaural audio for a given camera pose, our process begins by rendering a pair of RGB and depth images with the same camera pose using Eq. 3 and Eq. 4, respectively. We exploit a pre-trained encoder (e.g., ResNet-18 [43] pre-trained on ImageNet dataset [44]) to extract color and depth features from the RGB and depth images. The extracted features can serve as important indicators of the environment. Next, we develop an AV-Mapper, implemented as an MLP network, to project the color and depth features into a latent space. Our goal for the A-V Mapper is two-fold: (1) to extract meaningful embeddings while discarding unnecessary features, and (2) to align the features in the original space of the pre-trained image encoder with the space relevant to our specific problem. The output embedding of the AV-Mapper, denoted as $\mathbf{e} \in \mathcal{R}^c$, with $c$ representing the width of each linear layer in A-NeRF, is then added to the input of A-NeRF for controlling the audio synthesis. Ablation studies (Sec. 5.2) demonstrate the slight advantage of this fusion method over others.

### 4.4 Coordinate Transformation

Viewing direction $(\theta, \phi)$ in V-NeRF is expressed in an absolute coordinate system. This is a natural practice in visual space such that directions of two parallel light rays are expressed identically. However, this expression method in audio space is less suitable because the human perception of the sound direction is based on the relative direction to the sound source instead of the absolute direction. To address this limitation, we propose expressing the viewing direction of a camera pose relative to the sound source. This coordinate transformation encourages A-NeRF learning a sound source-centric acoustic field, enhancing spatial audio generation.

Given the 2D position $(x, y)$ and the direction $\theta$ of the camera, as well as the position of the sound source $(S_x, S_y)$, we obtain two direction vectors: $\mathbf{V}_1 = (S_x - x, S_y - y)$ and $\mathbf{V}_2 = (\cos(\theta), \sin(\theta))$. $\mathbf{V}_1$ represents the direction from the camera to the sound source, and $\mathbf{V}_2$ represents the camera's direction in the absolute coordinate system. By calculating the angle between $\mathbf{V}_1$ and $\mathbf{V}_2$, denoted as the relative direction $\theta' = \angle(\mathbf{V}_1, \mathbf{V}_2)$, we obtain the rotation angle relative to the sound source. This angle $\theta'$ allows different camera poses to share the same direction encoding if they face the sound source at the same angle.

After computing the relative angle $\theta'$, we choose learnable embeddings instead of positional encoding [16, 45] to project $\theta'$ into a high-frequency space. We use the embedding $\Theta \in \mathcal{R}^{4 \times c}$ to represent four discrete directions, namely, $0°$, $90°$, $180°$, and $270°$, where $4$ is the number of directions and $c$ is the width of A-NeRF. Given a relative angle $\theta' \in [0°, 360°)$, we linearly interpolate the direction embedding $\Theta' \in \mathcal{R}^c$ according to the angle between $\theta'$ and four discrete directions. We add the

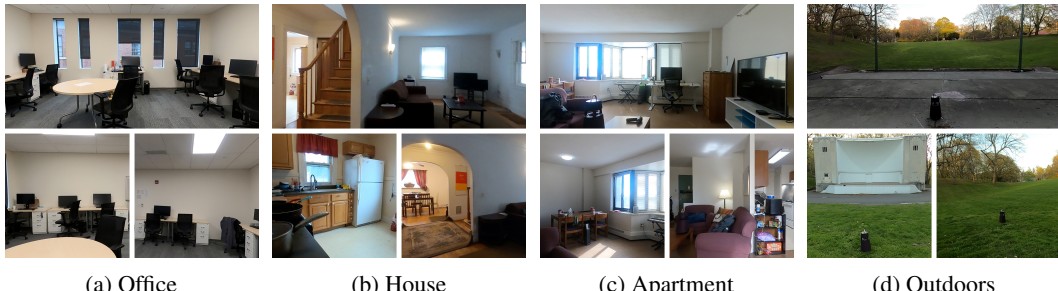

|  (a) Office | (b) House | (c) Apartment | (d) Outdoors |

Figure 5: Example scenes of RWAVS dataset. RWAVS dataset consists of diverse audio-visual scenes indoors and outdoors. Each environment possesses distinct acoustics and structures.

interpolated embedding $\theta'$ to the input of each linear layer in A-NeRF, thereby providing A-NeRF with direction information. Ablation studies (Sec. 5.2) show that this encoding method performs best.

### 4.5 Learning Objective

The loss function of V-NeRF is the same as [16]:

$$\mathcal{L}_V = ||C(\mathbf{r}) - \hat{C}(\mathbf{r})||^2 \ , \tag{6}$$

where $C(\mathbf{r})$ is the ground-truth color along the ray $\mathbf{r}$ and $\hat{C}(\mathbf{r})$ is the color rendered by V-NeRF. After training V-NeRF, we optimize A-NeRF and AV-Mapper together with the L2 loss function:

$$\mathcal{L}_A = ||s_m - \hat{s}_m||^2 + ||s_l - \hat{s}_l||^2 + ||s_r - \hat{s}_r||^2 \ , \tag{7}$$

where $s_{m,l,r}$ are the predicted magnitudes, $\hat{s}_{m,l,r}$ are the ground-truth magnitudes, and subscript $m, l, r$ are mixture, left, and right, respectively. The first term of $\mathcal{L}_A$ encourages A-NeRF to predict masks that represent spatial effects caused by distance, while the second and third term encourages A-NeRF to generate masks that capture the differences between two channels.

## 5 Experiments

### 5.1 Datasets

To the best of our knowledge, our method is the first NeRF-based system capable of synthesizing real-world videos with perceptually realistic binaural audios at arbitrary poses. However, existing datasets do not meet the specific requirements of our experiments, particularly in terms of simultaneously providing camera poses, high-quality binaural audios, and images. Therefore, we curated a high-quality audio-visual scene dataset (*real*) to address this gap and facilitate further research on this problem. Additionally, we utilize (*synthetic*) SoundSpaces dataset [4] to validate our method.

**(1) RWAVS Dataset.** We collected the Real-World Audio-Visual Scene (RWAVS) dataset to benchmark our method. In order to increase the diversity of our dataset, we recorded data across different scenarios. Fig. 5 shows the example scenarios we used for data recording, including both indoor and outdoor environments, which we believe represent most daily settings. RWAVS dataset comprises multimodal data, including camera poses, high-quality binaural audios, and videos. Unlike Replay-NVAS dataset [15], where the environment and the recording viewpoint are constant, RWAVS dataset contains various viewpoints in diverse environments. During data recording, we randomly moved around the environment while holding the device, capturing various acoustic and visual signals. RWAVS dataset encompasses all positions and directions (360°) within an environment.

In detail, we employed a 3Dio Free Space XLR binaural microphone for capturing high-quality stereo audio, a TASCAM DR-60DMKII for recording and storing audio, and a GoPro Max for capturing accompanying videos. Additionally, an LG XBOOM 360 omnidirectional speaker was used as the sound source. For each environment and sound source combination, we collected data ranging from 10 to 25 minutes, resulting in a total collection of 232 minutes (3.8 hours) of data from diverse environments with varying source positions.

We extract key frames at 1 fps from recorded videos and use COLMAP [46] to estimate the corresponding camera pose. Each key frame is accompanied by one-second binaural audio and one-second

Table 1: Comparison with state-of-the-art methods on RWAVS dataset.

| Methods | Office | | House | | Apartment | | Outdoors | | Overall | |
|---|---|---|---|---|---|---|---|---|---|---|
| | MAG | ENV | MAG | ENV | MAG | ENV | MAG | ENV | MAG | ENV |
| Mono-Mono | 9.269 | 0.411 | 11.889 | 0.424 | 15.120 | 0.474 | 13.957 | 0.470 | 12.559 | 0.445 |
| Mono-Energy | 1.536 | 0.142 | 4.307 | 0.180 | 3.911 | 0.192 | 1.634 | 0.127 | 2.847 | 0.160 |
| Stereo-Energy | 1.511 | 0.139 | 4.301 | 0.180 | 3.895 | 0.191 | 1.612 | 0.124 | 2.830 | 0.159 |
| INRAS [13] | 1.405 | 0.141 | 3.511 | 0.182 | 3.421 | 0.201 | 1.502 | 0.130 | 2.460 | 0.164 |
| NAF [12] | 1.244 | 0.137 | 3.259 | 0.178 | 3.345 | 0.193 | 1.284 | 0.121 | 2.283 | 0.157 |
| ViGAS [15] | 1.049 | 0.132 | 2.502 | 0.161 | 2.600 | 0.187 | 1.169 | 0.121 | 1.830 | 0.150 |
| Ours | **0.930** | **0.129** | **2.009** | **0.155** | **2.230** | **0.184** | **0.845** | **0.111** | **1.504** | **0.145** |

Table 2: Ablation studies. We break down AV-NeRF to analyze the contribution of each component. **Left:** the inclusion of AV-Mapper (AV) and Coordinate Transformation (CT), **Middle:** different multimodal fusion methods, and **Right:** different direction encoding methods.

| Methods | Overall | | Methods | Overall | | Methods | Overall | |
|---|---|---|---|---|---|---|---|---|
| | MAG | ENV | | MAG | ENV | | MAG | ENV |
| Baseline | 2.287 | 0.157 | Concat Input | 1.507 | 0.145 | Absolute Direction | 1.701 | 0.149 |
| Ours w/o AV | 1.791 | 0.150 | Add Input | **1.504** | **0.145** | Relative Direction | 1.508 | 0.145 |
| Ours w/o CT | 1.701 | 0.149 | Add All Layers | 1.505 | 0.145 | Relative Embedding | **1.504** | **0.145** |
| Ours | **1.504** | **0.145** | | | | | | |

source audio, forming a complete data sample. For audio clips with noticeable background noise, we perform noise suppression using Adobe Audition [47]. We split 80% data as training samples and the rest as validation samples. After pre-processing, we obtain 9850 and 2469 samples for training and validation, respectively. This dataset is challenging because of the diverse environments and various camera poses. We will release this dataset to the research community.

**(2) SoundSpaces Dataset.** While RWAVS offers realistic training samples, its realism restricts its scale because it is time-consuming to record high-quality multimodal data in the real world. Therefore, we use the synthetic SoundSpaces dataset to augment our experiments. To evaluate our method on SoundSpaces dataset, we modify AV-NeRF to estimate impulse responses instead of the acoustic mask while keeping all other components intact. We follow NAF [12] selecting six representative indoor scenes, consisting of two single rooms with rectangular walls, two single rooms with non-rectangular walls, and two multi-room layouts. In each scene, SoundSpaces dataset provides an extensive collection of impulse response signals for sound source and sound receiver pairs, which are densely sampled from a 2D room grid. Each pair includes four discrete head orientations (0°, 90°, 180°, and 270°), and each orientation is associated with two-channel binaural RIRs. We render RGB and depth images for each sound receiver pose using Habitat-Sim simulator [48, 49]. We maintain the same training/test split as NAF, allocating 90% data for training and 10% data for testing.

## 5.2 Results on RWAVS Dataset

**Comparison with State-of-the-art.** We compare AV-NeRF with the following baselines: (1) Mono-Mono duplicates the source audio $a_s$ twice to generate a fake binaural audio without modifying the source audio; (2) Mono-Energy assumes that the average energy of the target audio $a_t$ is known, scales the energy of the input audio to match the target, and duplicates the scaled audio to generate a stereo audio; (3) Stereo-Energy assumes that the energy of the two channels of the target audio $a_t$ is known, separately scales the energy of the input audio to match the target, and combines the two scaled channels to generate a stereo audio; (4) IRNAS [13] learns representing audio scenes by disentangling scene's geometry features with implicit neural fields, and we adapt INRAS to predict wave masks on RWAVS dataset; (5) NAF [12] designs local feature grids and an implicit decoder to capture the sound propagation in a physical scene, and we modify NAF to predict magnitude masks on RWAVS dataset; (6) ViGAS [15] achieves novel-view acoustic synthesis by analyzing audio-visual cues from source viewpoints. We select magnitude distance (MAG) [29], which measures the audio quality in the time-frequency domain, and envelope distance (ENV) [30], which measures the audio quality in the time domain, to evaluate various methods. Please refer to the supplementary material for implementation details.

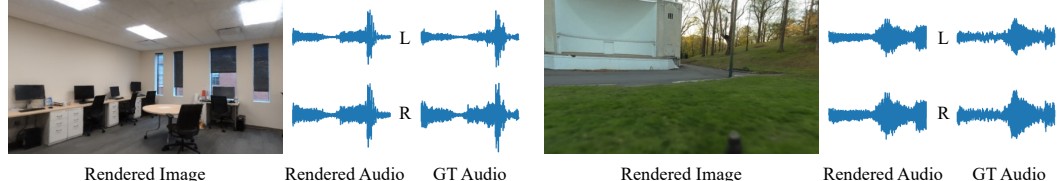

Rendered Image    Rendered Audio    GT Audio      Rendered Image    Rendered Audio    GT Audio

Figure 6: Visualization of synthesized audio-visual scene. We present the rendered image, synthesized binaural audio, and ground-truth audio.

AV-NeRF outperforms all baselines across different environments, including office, house, apartment, and outdoors, by a significant margin (Table 1). AV-NeRF outruns INRAS with 0.956 on the overall MAG metric (39% relative improvement) and 0.019 on the average ENV metric (11.6%). AV-NeRF surpasses NAF with 0.779 on MAG metric (34%) and 0.012 on ENV metric (8%). Our approach is better than ViGAS in terms of both MAG (1.504 compared to ViGAS's 1.830) and ENV (0.145 compared to ViGAS's 0.150).

**Ablation Studies.** We conduct ablation studies on AV-NeRF to analyze the contribution of different components. We combine A-NeRF and V-NeRF as our baseline, which does not contain AV-Mapper (Sec. 4.3) or coordinate transformation (Sec. 4.4). The ablation results are as follows: (1) AV-Mapper (AV) and coordinate transformation (CT) play important roles in learning audio scenes (Table 2 left). The exclusion of either component degrades the generation quality; (2) adding visual information to the input of A-NeRF is the most effective multimodal fusion method compared with concatenation and adding visual information to all layers of A-NeRF (Table 2 middle); (3) using embeddings to represent relative angles outperforms applying positional encoding to either absolute or relative angles (Table 2 right). "Absolute Direction" represents applying positional encoding to the absolute angle, "Relative Direction" means transforming the relative angle with the positional encoding, and "Relative Embedding" is the embedding method.

**Visualization.** We visualize the synthesized audio-visual scenes in Fig. 6 to intuitively assess the generation quality of our model. AV-NeRF can synthesize realistic binaural audios that have the same signal envelope and channel difference as the ground-truth audios.

### 5.3 Results on SoundSpaces Dataset

We compare AV-NeRF with traditional audio coding methods [50, 51] and advanced learning-based neural field methods [12, 13] using T60, C50, and EDT metrics [13]. Please refer to our supplementary material for implementation details. Table 3 shows that AV-NeRF outruns both traditional and advanced methods, achieving 21% relative improvement on T60 metric compared with the previous state-of-the-art method INRAS, 5% on C50, and 16% on EDT.

Table 3: Comparison with state-of-the-art. We report the performance on the SonudSpaces dataset using T60, C50, and EDT metrics. The lower score indicates a higher RIR generation quality. Opus is an open audio codec [50], and AAC is a multi-channel audio coding standard [51].

| Methods | T60 (%) ↓ | C50 (dB) ↓ | EDT (sec) ↓ |
|---|---|---|---|
| Opus-nearest | 10.10 | 3.58 | 0.115 |
| Opus-linear | 8.64 | 3.13 | 0.097 |
| AAC-nearest | 9.35 | 1.67 | 0.059 |
| AAC-linear | 7.88 | 1.68 | 0.057 |
| NAF [12] | 3.18 | 1.06 | 0.031 |
| INRAS [13] | 3.14 | 0.60 | 0.019 |
| Ours | **2.47** | **0.57** | **0.016** |

## 6 Discussion

In this work, we propose a first-of-its-kind NeRF system capable of synthesizing real-world audio-visual scenes. Our model can generate audios with rich spatial information at novel camera poses. We demonstrate the effectiveness of our method on real RWAVS and synthetic SoundSpaces datasets.

**Limitation.** Firstly, we currently focus on static scenes with a single fixed sound source in our study. However, it is worth exploring the challenge of learning implicit neural representations for audio-visual scenes with multiple dynamic sound sources. Second, while our model successfully generates audio-visual scenes, it does not account for reverberation effects present in the real world. Reverberation is essential for humans to perceive scene size and structure. Third, similar to the original NeRF, AV-NeRF must represent and render each scene separately. Developing a neural field

that can either learn to represent all scenes or transfer knowledge to new environments is a crucial problem, particularly for industrial applications.

**Broader Impact.** Although AV-NeRF is designed to synthesize audio-visual scenes, it is important to acknowledge its potential application in generating scenes involving artificial humans. The misuse of AV-NeRF could lead to the production of deceptive and misleading media.

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

# A    Additional Visualization Results

We present additional results in Figure 7 that demonstrate AV-NeRF's capability of generating distance-aware auditory effects. For each scene, we select a fixed source audio and synthesize new binaural audio at varying positions and distances from the sound source. The first row displays the rendered images, while the second row showcases the corresponding rendered audio. As illustrated in the figure, the amplitude of the rendered audio diminishes as the distance between the source and the camera increases, and conversely, it increases as the camera approaches the source.

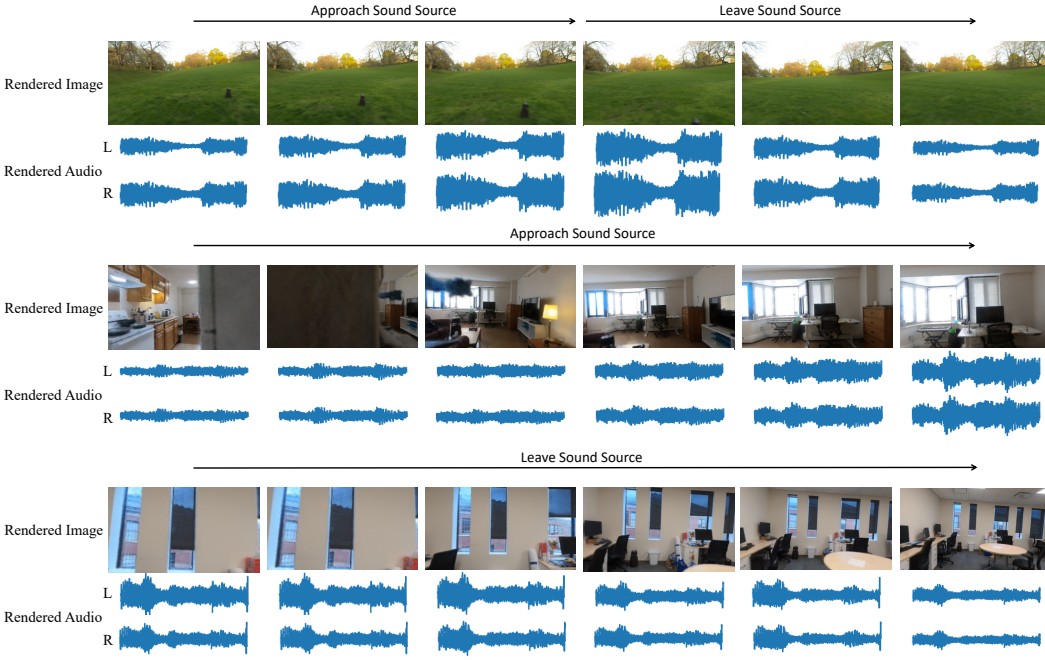

Figure 7: Distance-aware audio rendering. We show some example scenes where AV-NeRF can successfully synthesize consistent binaural audio with the camera movements.

# B    Failure Cases

We include several failure cases in Figure 8, where we present the rendered images, synthesized binaural audio, and ground-truth audio. AV-NeRF makes wrong acoustic predictions when the audio-visual scene involves noticeable noise or the AV-Mapper can not extract reliable material and geometry information from the visual space.

Because we work for **real-world** audio-visual learning, there inevitably exist various types of noise in a scene, e.g., sounds from refrigerators, air conditioning, wind, or even workers make noise when collecting data. These ambient noises are recorded by the microphone and are present in the ground-truth binaural audio, which hinders AV-NeRF from accurately learning the acoustic field. In Figure 8 (a), the ground-truth audio (used for training) marked with red boxes contains noticeable noise, resulting in inaccurate audio rendering.

AV-NeRF relies on the AV-Mapper to extract reliable material and geometry information from input images, enabling a comprehensive understanding of the audio-visual environment. However, in cases where the AV-Mapper fails to extract meaningful information from certain images (e.g., images of plain walls or whiteboards), AV-NeRF predicts erroneous acoustic masks. In Figure 8 (b), the rendered audio marked with black boxes displays inconsistent waveforms compared to the ground-truth audio, indicating a model failure due to meaningless material and geometry information.

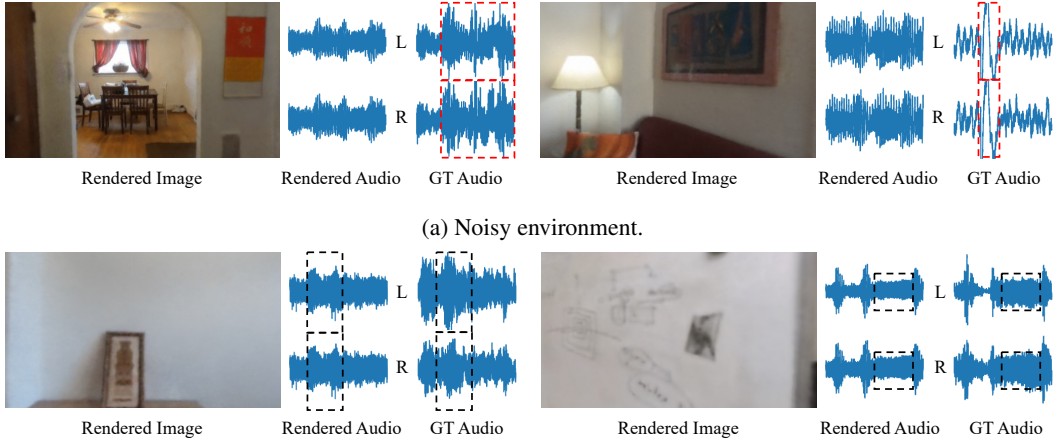

Rendered Image  Rendered Audio GT Audio   Rendered Image  Rendered Audio GT Audio

(a) Noisy environment.

Rendered Image  Rendered Audio GT Audio   Rendered Image  Rendered Audio GT Audio

(b) Meaningless visual input.

Figure 8: Failure cases. We present failure cases in which AV-NeRF makes wrong acoustic predictions. For each case, we show the rendered image, the rendered audio, and the recorded ground-truth audio.

## C Rationality of AV-Mapper

In our paper, we leverage RGB and depth images as *implicit* indicators of environmental material properties and geometry. While these images enable the model to perceive the environment implicitly, it is important to note that achieving a precise one-to-one mapping from images to material properties or geometry is not guaranteed for AV-Mapper. Certain corner cases, such as a black desk and object occlusion, can lead to the failure of AV-Mapper.

Nevertheless, existing studies have demonstrated the feasibility of inferring material and geometry information from images. For instance, back in 2002, Varma and Zisserman [52] proposed a filter-based approach for material classification in images. More recently, in 2017, Zhang et al. [53] successfully recognized material and shape attributes using visual data. Moreover, there is a body of audio-visual research indicating that extracted geometry and material information via ResNet networks can contribute to audio synthesis [54, 55]. Our paper further supports this concept. As illustrated in Table 2 (left), the inclusion of AV-Mapper in AV-NeRF can improve the MAG score by 16% from 1.791 to 1.504 and enhance the ENV score by 3% from 0.150 to 0.145.

## D Necessity of Distance and Direction Coordinates

Table 4: Necessity of distance and direction information.

| Methods | | Overall | |
|---|---|---|---|
| Position | Direction | MAG | ENV |
| | | 1.822 | 0.156 |
| | ✓ | 1.817 | 0.155 |
| ✓ | | 1.688 | 0.148 |
| ✓ | ✓ | **1.504** | **0.145** |

To validate the necessity of distance and direction coordinates when modeling audio fields, we conduct additional ablation studies. We intentionally exclude positional and directional coordinates when representing an auditory scene and evaluate the performance without these inputs. The results are presented in Table 4. AV-NeRF achieves a MAG metric score of 1.504 and an ENV metric score of 0.145 when both position and direction information are utilized. Removing either positional or directional input leads to a performance drop between 2% to 21%. In the absence of both positional and directional inputs, performance degrades further to 1.822 on the MAG metric and 0.156 on

the ENV metric. These outcomes distinctly affirm the indispensable role of distance and direction coordinates in acoustic modeling.

# E   Multiple Sound Sources

Given that scenes with a single sound source may not fully represent real-world complexities, we extend both the RWAVS dataset and the AV-NeRF model to support multi-source scenes.

Specifically, we collect two multi-source scenes, adhering to the recording settings outlined in Section 5.1. The only modification we make is placing two sound sources instead of a single one in the environment, thus creating multiple sound sources. We extend AV-NeRF by stacking multiple equal A-NeRF modules to support the parameterization of multi-source acoustic fields. AV-NeRF generates acoustic masks for each sound source separately. We conduct experiments in these multi-source scenes and present the performance of AV-NeRF and other baselines in Table 5. As presented in the table, AV-NeRF outperforms other baselines in both MAG (0.282) and ENV (0.063) metrics. The experimental results clearly demonstrate the proposed method's capability to effectively handle multiple sound sources.

Table 5: Comparison with state-of-the-art methods on multi-source scenes.

| Methods | Multi-source | |
| --- | --- | --- |
| | MAG | ENV |
| Mono-Mono | 1.949 | 0.172 |
| Mono-Energy | 0.533 | 0.075 |
| Stereo-Energy | 0.527 | 0.073 |
| INRAS [13] | 0.472 | 0.078 |
| NAF [12] | 0.401 | 0.080 |
| Ours | **0.282** | **0.063** |

# F   Architectures

**A-NeRF.** A-NeRF consists of two Multilayer Perceptrons (MLPs), each comprising four linear layers with an additional residual connection. The width of each linear layer, denoted as $c$, is set to 128 for the RWAVS dataset and 256 for the SoundSpaces dataset. In A-NeRF, all linear layers are followed by ReLU activation layers, except for the last layer, where the ReLU activation is replaced with the Sigmoid function. The first MLP takes the listener's position $(x, y)$ and the frequency $f \in [0, F]$ as input, where $F$ represents the number of frequency bins. It predicts a mixture mask $m_m \in \mathcal{R}$ for the given frequency $f$ and generates a feature vector with $c$ channels. Prior to feeding them into the MLP, we apply positional encoding to the listener's position $(x, y)$ and the frequency $f$. We set the maximum frequency used for positional encoding as 10.

Then, we adopt relative transformation (Sec. 4.4) to project the listener's direction $\theta$ into a high-frequency space. We concatenate the transformed listener's direction and the feature vector, and feed it into the second MLP. The second MLP is appended with a Sigmoid layer and a scaling layer, ensuring that the difference mask $m_d$ estimated by the second MLP falls within the range of $[-1, 1]$. For each frequency query $f$, A-NeRF estimates two masks: $m_m$ and $m_d$, both of which are scalars. We iterate over all frequencies $f \in [0, F]$ to obtain the complete masks $\mathbf{m}_m$ and $\mathbf{m}_d$. After computing the masks, we synthesize the target audio $a_t$ according to the procedure discussed in Sec. 4.2.

**V-NeRF.** We utilize the nerfacto model provided by nerf-studio [20] as the V-NeRF. This model combines several well-established and successful methods, including camera pose refinement [56, 57], image appearance conditioning [24], hash encoding, and proposal sampling [58]. Due to its robust and effective performance on real-world data, we utilize the default settings of the nerfacto model without making any architectural modifications. For more detailed information regarding the architecture of V-NeRF, please refer to the documentation provided by nerf-studio.

**AV-Mapper.** For each camera pose, we render both RGB and depth images using V-NeRF. We resize images to $256 \times 256$ and center-crop to $224 \times 224$ prior to feeding them into a frozen ResNet-18 [43]

image encoder pre-trained on ImageNet-1K dataset [44]. ResNet-18 embeds the input image as a 512-dimension feature vector. We concatenate the RGB and the depth feature vectors, and input them into the AV-Mapper to learn environmental knowledge of the sound acoustics. AV-Mapper projects the input feature vectors to a latent embedding of $c$ channels. We implement the AV-Mapper as a 3-layer MLP, with each intermediate linear layer followed by a ReLU activation function.

## G    Implementation Details

**RWAVS Dataset.** We implement our method using the PyTorch framework [59]. We employ Adam optimizer [60] with $\beta_1 = 0.9$ and $\beta_2 = 0.999$ for model optimization. The initial learning rate is set to $5e-4$ and exponentially decreased to $5e-6$. We train the model for $100$ epochs with a batch size of $32$.

Before feeding the camera position $(x, y)$ to A-NeRF, we normalize it within the range $[-1, 1] \times [-1, 1]$ and apply positional encoding [16]. Additionally, we resample all audios to a frequency of $22050$ Hz and utilize the Short-Time Fourier Transform (STFT) to convert waveform audios into the time-frequency domain. For this transformation, we set the number of ffts as $512$, the window length as $512$, and the hop length as $128$. A Hanning window is applied during the process. Finally, we compute both the magnitude and the phase from the spectrogram.

We use magnitude distance (MAG) [29] and envelope distance (ENV) [30] as evaluation metrics for audio quality. The MAG metric quantifies audio quality in the time-frequency domain and is defined as follows:

$$\text{MAG}(\mathbf{m}_{\text{prd}}, \mathbf{m}_{\text{gt}}) = ||\mathbf{m}_{\text{prd}} - \mathbf{m}_{\text{gt}}||^2 \ , \tag{8}$$

where $\mathbf{m}_{\text{prd}}$ is the predicted magnitude, and $\mathbf{m}_{\text{gt}}$ is the ground-truth magnitude. ENV metric that measures the audio quality in the time domain is formatted as:

$$\text{ENV}(a_{\text{prd}}, a_{\text{gt}}) = ||\text{hilbert}(a_{\text{prd}}) - \text{hilbert}(a_{\text{gt}})||^2 \ , \tag{9}$$

where $a_{\text{prd}}$ is the predicted audio, $a_{\text{gt}}$ is the ground-truth audio, and $\text{hilbert}$ is the Hilbert transformation function [61].

**SoundSpaces Dataset.** Our model is trained on the SoundSpaces dataset using the same training settings as RWAVS dataset. We resample the impulse responses to $22050$ Hz following INRAS [13]. The 2D position is normalized to $[-1, 1] \times [-1, 1]$ prior to positional encoding.

We tailor A-NeRF for impulse response prediction with some minor modifications: (1) The input frequency query $f$ is replaced by a time query $t \in [0, T]$, where $T$ represents the length of an impulse response signal; (2) the first MLP only generates a feature vector while discarding the mixture mask $\mathbf{m}_m$; (3) the second MLP predicts impulse response signals instead of difference mask $\mathbf{m}_d$.

Since the generated impulse responses are in the time domain, we employ STFT to convert them into the time-frequency domain and calculate their magnitudes. We utilize an STFT configuration with $512$ FFTs, a sliding window width of $512$, a hop stride of $128$, and a Hanning window. We supervise the model training using the L2 distance between the ground-truth magnitudes and the predicted magnitudes.

For performance evaluation, we choose three metrics: T60, C50, and EDT [13]. T60 characterizes the reverberation effects in an audio signal by measuring the time it takes for the audio's energy to attenuate by 60 dB. The T60 distance is calculated as follows:

$$\text{T60}(a_{\text{prd}}, a_{\text{gt}}) = \frac{|\text{T60}(a_{\text{prd}}) - \text{T60}(a_{\text{gt}})|}{\text{T60}(a_{\text{gt}})} \ , \tag{10}$$

where $a_{\text{prd}}$ and $a_{\text{gt}}$ are the predicted and ground-truth impulse responses, respectively. C50 quantifies the energy ratio between early reflections and late reverberation, allowing it to represent the clarity and loudness of the audio. We format the C50 distance as:

$$\text{C50}(a_{\text{prd}}, a_{\text{gt}}) = |\text{C50}(a_{\text{prd}}) - \text{C50}(a_{\text{gt}})| \ . \tag{11}$$

The EDT metric shares similarities with T60 but places greater emphasis on capturing the early reflections of impulse responses. The EDT distance is defined as follows:

$$\text{EDT}(a_{\text{prd}}, a_{\text{gt}}) = |\text{EDT}(a_{\text{prd}}) - \text{EDT}(a_{\text{gt}})| \ . \tag{12}$$

With these three metrics, we can evaluate the generation quality of impulse responses from different aspects, including clarity, energy, and reverberation.

## H  Setup of RWAVS Dataset

**Recording Devices.** We have assembled a recording system, as depicted in Fig. 9, to capture high-quality audio-visual scenes in real-world environments. Our system comprises a 3Dio Free Space XLR binaural microphone for capturing stereo audio, a TASCAM DR-60DMKII for recording and storing audio, and a GoPro Max for capturing accompanying videos.

This system is portable, allowing us to position it flexibly and capture scenes from different camera poses. In addition, we utilized an LG XBOOM 360 omnidirectional speaker to serve as a sound source, which plays music repeatedly. Figure 5 in the main paper illustrates the setup used to record data in four distinct environments: office, house, apartment, and outdoors. Within each environment, we positioned the speaker at multiple locations to capture diverse acoustic effects. Each combination of environment and sound source represents an audio-visual scene. We collected data ranging from 10 to 25 minutes for each scene, resulting in a total collection of 232 minutes (3.8 hours) of diverse data, encompassing various environments and source positions.

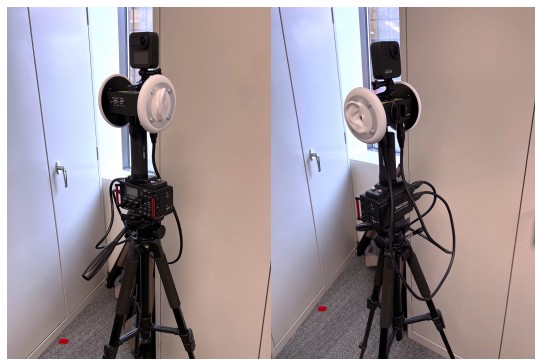

Figure 9: Recording system. It comprises a professional binaural microphone, a sports camera, and a recorder.

**Example Scenes.** In Fig.10, we present four example scenes from RWAVS dataset along with the corresponding camera pose distributions. The first column showcases images of the example scenes. The second column displays the camera poses used for video recording: the black dot represents the sound source, and each blue triangle represents a camera pose. We normalize all camera poses to the range of $[-1, 1]$ and visualize them in an x-y plane from a top-down view. The third column contains 2D density heatmaps, which illustrate the distribution of camera poses in each unit area: each pixel represents a unit area and its color shows the number of camera poses in this area. As shown in the figure, RWAVS dataset encompasses densely covered camera poses for each environment. We also analyze the distribution of camera directions (shown in the last column of Fig.10). We present the direction distribution in a polar coordinate system with the angle representing the viewing direction and the radius meaning the number of camera poses in this region. RWAVS dataset consists of various viewpoints that approximately cover a 360°range of viewing directions. In summary, the RWAVS dataset comprises diverse environments with a wide range of camera poses.

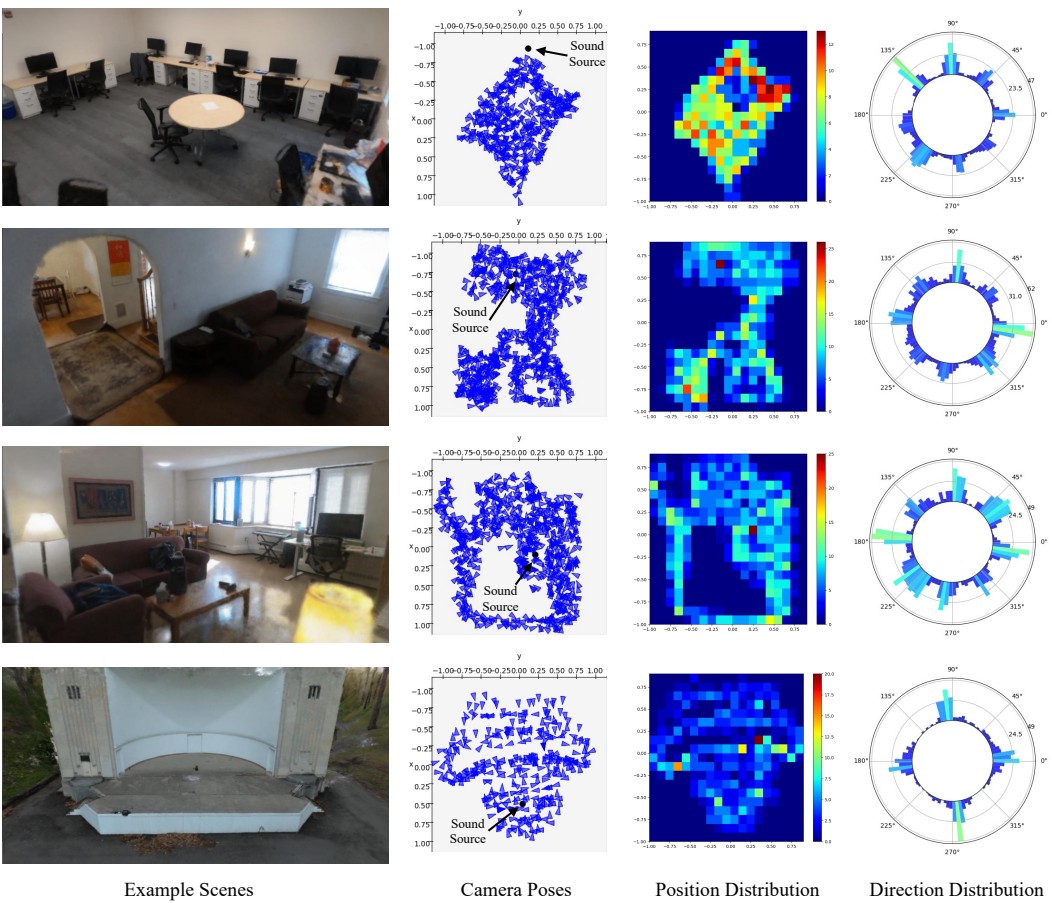

| Example Scenes | Camera Poses | Position Distribution | Direction Distribution |

Figure 10: Example scenes. We present several example scenes along with their corresponding camera pose distributions. We display the position density heatmap and the direction distribution map. RWAVS dataset is composed of diverse environments with various camera poses.

