# OpenReview forum: "AV-NeRF: Learning Neural Fields for Real-World Audio-Visual Scene Synthesis"
_NeurIPS.cc/2023/Conference — NeurIPS 2023 poster_

### Official Review · Reviewer_4psZ · 2023-07-02

**Soundness:** 3 good
**Presentation:** 3 good
**Contribution:** 3 good
**Rating:** 6
**Confidence:** 4

**Summary:**

The paper introduces a new task called "real-world audio-visual scene synthesis", which involves synthesizing new videos with spatial audios along arbitrary novel camera trajectories in a scene. The proposed NeRF-based approach integrates a prior knowledge of audio into audio generation and associates it with the 3D visual environment. The proposed model is able to learn a sound source-centric acoustic field with the introduction of the coordinate transformation module that expresses a view direction relative to the sound source. Additionally, a high-quality Real-World Audio-Visual Scene (RWAVS) dataset is demonstrated along with the model.

**Strengths:**

1.	The paper presents an interesting new task that associates the audio with 3D geometry and material properties of scenes and the video demo demonstrates the effectiveness of the proposed method.
2.	The paper compares the proposed method with similar methods on both the RWAVS dataset and the SoundSpaces dataset and surpasses the previous state-of-the-art method.
3.	The method proposes a novel dataset that can be further utilized in similar tasks.


**Weaknesses:**

1.	The novelty of the proposed network seems to be limited. To tackle the new task, AV-NeRF combines an audio-NeRF and visual-NeRF and introduces an AV-mapper and the novelty mostly lies in the AV-mapper based on my current understanding.

2.	The paper introduces a new task called "real-world audio-visual scene synthesis”.  However, as mentioned in the paper, this work currently focuses on static scenes with a single fixed sound source, which is a bit far from the word “real-world” since in real-world, there can be much more sound sources.


**Questions:**

1.	While the method is spatially-aware, the method seems to be less effective when the distance of the sound source is changing from the demonstration of the video, could you provide more results?

2.	The sound source demonstrated in the video is the same, could the sound source be changed, or the sound-source device has certain requirements? If so, the proposed method can be limited.

3.	As mentioned in the weaknesses, please clarify the novelty of the proposed method again.


**Limitations:**

As mentioned in the paper, this work currently focuses on static scenes with a single fixed sound source. To better suit the word “real-world”, the method needs to be improved and adapted to multiple sound-source situations.

---

> ### Author Rebuttal · Authors · 2023-08-09
>
> We thank the reviewer for your insightful comments. Below are our responses to specific questions.
>
> ### Weakness
> **W1: Novelty of AV-NeRF.** In addition to AV-Mapper, our paper introduces two essential components -- the innovative Audio-NeRF architecture and a novel coordinate transformation mechanism.
>
> We would like to clarify that it is non-trivial to extend NeRF to the audio domain.
> Our A-NeRF pipeline is carefully designed to capture the influence of the receiver's position and direction on sound perception. It consists of two separate Multi-layer Perceptrons (MLP) to learn the acoustic field, with the first MLP parameterizing the energy attenuation with regard to the distance between the sound source and receiver and the second one modeling the channel difference of stereo sound caused by the receiver's viewing direction. By disentangling the acoustic field with these two modules, A-NeRF enables distance-sensitive and direction-aware audio synthesis.
>
> Another contribution of our work is the proposed coordinate transformation. By studying the nature of human's perception of the sound source -- human perception of the sound direction is based on the relative direction to the sound source instead of the absolute direction -- we propose replacing the absolute coordinate system, which is commonly used in NeRF, with a relative coordinate system. By expressing the viewing direction of a camera pose relative to the sound source, we encourage A-NeRF to learn a sound source-centric acoustic field and establish a strong correspondence between the viewing direction and the spatial audio.
>
> **W2: Multiple sound sources.** We agree that there are multiple sound sources in real-world scenarios, and the initial setting of this work was simplified. To address this limitation, we extend both the RWAVS dataset and the AV-NeRF model to support multi-source scenes. Specifically, we collect two multi-source scenes, adhering to the recording settings outlined in Section 5.1. The only modification we make is placing two sound sources instead of a single one in the environment, thus creating multiple sound sources. We extend AV-NeRF by stacking multiple equal A-NeRF modules to support the parameterization of multi-source acoustic fields. AV-NeRF generates acoustic masks for each sound source separately. We conduct experiments in these multi-source scenes and present the performance of AV-NeRF and other baselines in the table below.
>
> |Methods|Multi-source (MAG) | Multi-source (ENV)|
> |:-:|:-:|:-:|
> |Mono-Mono | 1.949 | 0.172 |
> Mono-Energy | 0.533 | 0.075 |
> Stereo-Energy | 0.527 | 0.073 |
> INRAS | 0.472 | 0.078 |
> NAF | 0.401 | 0.080 |
> Ours | **0.282** | **0.063** |
>
> As presented in the table, AV-NeRF outperforms other baselines in both MAG (0.282) and ENV (0.063) metrics. The experimental results clearly demonstrate the proposed method's capability to effectively handle multiple sound sources. In our revision, we will augment the RWAVS dataset with multi-source scenes and include the corresponding results in the main paper.
>
> ### Question
> **Q1: Distance effects.** In response to the reviewer's question, we present additional results in Figure 2 (please refer to
> the attached PDF file in the global response) that demonstrate AV-NeRF's capability to generate distance-aware auditory effects. For each scene, we select a fixed source audio and synthesize new binaural audio at varying positions and distances from the sound source. The first row displays the rendered images, while the second row showcases the corresponding rendered audio. As illustrated in the figure, the amplitude of the rendered audio diminishes as the distance between the source and the camera increases, and conversely, it increases as the camera approaches the source.
>
> **Q2: Sound source.** There are no specific requirements for the sound source device. It can be any object that emits sound, including phones, laptops, instruments, or even humans. In our case, we utilize a speaker as the sound source to simplify the data recording process.
>
> **Q3: Please refer to W1.**
>
> ### Limitation
> **L1: Please refer to W2.**

---

> > ### Comment · Reviewer_4psZ · 2023-08-13
> > **Comment**
> >
> > Thank you for the clarification and the efforts you put into the rebuttal. All the questions are responded and the novelties are clarified in the response. The spatial awareness is demonstrated in the rebuttal and the effectiveness is proved. I would incline to raise my rating.

---

> > > ### Author Response · Authors · 2023-08-13
> > >
> > > We are grateful for the reviewer's encouraging comments. We will incorporate multi-source results and spatial awareness figures into our revised paper.

---

### Official Review · Reviewer_iWmP · 2023-07-05

**Soundness:** 3 good
**Presentation:** 4 excellent
**Contribution:** 4 excellent
**Rating:** 8
**Confidence:** 4

**Summary:**

The paper introduces a novel task called real-world audio-visual scene synthesis to generate novel views and spatial audio for any camera trajectories. The system contains a visual NeRF to synthesize novel view images, an audio NeRF to generate acoustic masks, and an audio-visual mapper to enable the visual information to enhance the A-NeRF. The authors also collect a new real-world dataset to demonstrate the task's importance and for benchmarking. Experiments on synthetic and real datasets show the effectiveness of the proposed method.

**Strengths:**

1. The task of audio-visual scene synthesis in the real world is essential. To my best knowledge, it is the first work that aims to generate both video and spatial audio in a real environment.

2. Multimodal data collection for the real scene is challenging and requires lots of effort. I could see the new real data would be very useful for future research.

3. Each component in the model pipeline is well-motivated and makes sense.

4. The paper is generally well-written and easy to follow.

5. Extensive experiments and ablation studies are done on real and synthetic data to show the effectiveness of the proposed method.

**Weaknesses:**

1. As the authors mention in the limitation, the current approach mainly works for static scenes with a single fixed sound source. However, as the initial exploration, the current setup makes sense.

2. It is often challenging to interpret and understand the quantitative results of generated audio. While the proposed method outperforms baselines across all quantitative metrics. If possible, I would suggest authors add human subjective evaluations in the future to see the perceptual difference.

3. As a new benchmark, it would be great to show some failure cases for the current method to indicate potential improvement in the future.

**Questions:**

1. If I understand correctly, the source mono audio is fixed for each scene in the RWAVS dataset. Have you tried to use another source audio during the inference time, and does it sound as good as before?

2. Following the previous question, I guess one of the main differences between the real world and simulated data is that in the simulation platform, we could obtain dense impulse responses in arbitrary positions, but in the real world, it is obviously easier to play source audio and collect data along a trajectory continuously. However, how could we effectively validate the correctness at the inference stage if we are using new source audio without spatial audio ground truth?

3. Since the goal is to generate video frames and spatial audio, the current A-V mapper aims to extract useful information from visuals to help A-NeRF. Is it also possible to use it to extract useful information from the audio side to enhance the synthesized visual frame?

---

> ### Author Rebuttal · Authors · 2023-08-09
>
> We sincerely thank the reviewer for your constructive comments and greatly appreciate your strong positive acknowledgment of our work. Below, we address each specific question.
>
> ### Weakness
> **W1: Single sound source.** In our paper, we initially focused on a simplified scenario with a single sound source, acknowledging that it may not fully represent real-world complexities. We appreciate the reviewer's understanding that "as the initial exploration, the current setting makes sense." To address this limitation, we extend both the RWAVS dataset and the AV-NeRF model to support multi-source scenes. Specifically, we collect two multi-source scenes, adhering to the recording settings outlined in Section 5.1. The only modification we make is placing two sound sources instead of a single one in the environment, thus creating multiple sound sources. We extend AV-NeRF by stacking multiple equal A-NeRF modules to support the parameterization of multi-source acoustic fields. AV-NeRF generates acoustic masks for each sound source separately. We conduct experiments in these multi-source scenes and present the performance of AV-NeRF and other baselines in the table below.
>
> |Methods|Multi-source (MAG) | Multi-source (ENV)|
> |:-:|:-:|:-:|
> |Mono-Mono | 1.949 | 0.172 |
> Mono-Energy | 0.533 | 0.075 |
> Stereo-Energy | 0.527 | 0.073 |
> INRAS | 0.472 | 0.078 |
> NAF | 0.401 | 0.080 |
> Ours | **0.282** | **0.063** |
>
> As presented in the table, AV-NeRF outperforms other baselines in both MAG (0.282) and ENV (0.063) metrics. The experimental results clearly demonstrate the proposed method's capability to effectively handle multiple sound sources. In our revision, we will augment the RWAVS dataset with multi-source scenes and include the corresponding results in the main paper.
>
> **W2: Human subjective evaluation.** We agree with the reviewer's valid concern regarding the interpretability and comprehension of the quantitative results for the generated audio. To address this, we plan to conduct a human subjective evaluation in the future to perceptually assess the quality of the generated content.
>
> **W3: Failure cases.** As suggested by the reviewer, we include failure cases in Figure 1 (please refer to the attached PDF file in the global response). In the figure, we present the rendered images, synthesized binaural audio, and ground-truth audio. AV-NeRF makes wrong acoustic predictions when the audio-visual scene involves noticeable noise or the AV-Mapper can not extract reliable material and geometry information from the visual space.
>
> 1. Because we work for **real-world** audio-visual learning, there inevitably exist various types of noise in a scene, e.g., sounds from refrigerators, air conditioning, wind, or even workers make noise when collecting data. These ambient noises are recorded by the microphone and are present in the ground-truth binaural audio, which hinders AV-NeRF from accurately learning the acoustic field. In the Figure 1 (a), the ground-truth audio (used for training) marked with red boxes contains noticeable noise, resulting in inaccurate audio rendering.
>
> 2. AV-NeRF relies on the AV-Mapper to extract reliable material and geometry information from input images, enabling a comprehensive understanding of the audio-visual environment. However, in cases where the AV-Mapper fails to extract meaningful information from certain images (e.g., images of plain walls or whiteboards), AV-NeRF predicts erroneous acoustic masks. In Figure 1 (b), the rendered audio marked with black boxes displays inconsistent waveforms compared to the ground-truth audio.
>
> ### Question
> **Q1: Replace source audio.** For quantitative evaluation, we use a fixed source audio input for AV-NeRF, which guarantees that the source audio and recorded ground-truth audio share the same auditory content. However, during the general inference process, AV-NeRF can accept any source audio of interest, allowing us to generate binaural audio with rich acoustic properties. For instance, we collect an outdoor scene in the RWAVS dataset using a music song as the source audio, and train AV-NeRF with both the source music and the recorded music. After completing the training, we feed AV-NeRF with arbitrary source audio to synthesize new binaural audio. In our demo videos, we replaced the music with human speech and successfully synthesized coherent binaural sounds. The generated speech audio demonstrated consistency with the camera movements acoustically.
>
> **Q2: Unavailable ground-truth audio.** We can synthesize pseudo ground-truth audio to sidestep the unavailability of ground-truth audio when we use new source audio to evaluate the correctness of an approach. Although ground-truth audio is not directly available for these new samples, we have access to the original source audio clip $a_s$ and the binaural audio clip $a_t$ from the dataset. Leveraging a deep neural network [1], we can estimate the corresponding impulse response from $a_s$ and $a_t$. By convolving the estimated impulse response with the new source audio, we synthesize the pseudo ground-truth audio. This approach effectively provides us with ground-truth binaural audio for performance evaluation.
>
> **Q3: Enhance visual synthesis.** We agree with the reviewer's suggestion that extracting valuable auditory information can aid in visual learning. As shown in [2], acoustic information can enhance the quality of visual reconstruction. Moreover, [3] demonstrates that acoustic features extracted from sound echoes can assist in spatial perception tasks, such as monocular depth estimation and surface normal estimation. It would be very interesting to explore audio-assisted vision learning in the future.
>
> Please refer to the global rebuttal for reference.

---

> > ### Comment · Reviewer_iWmP · 2023-08-17
> > **Post Rebuttal**
> >
> > Thank the authors for the detailed responses. My questions are well addressed. I will keep my score and recommend accepting the paper.

---

> > > ### Author Response · Authors · 2023-08-18
> > >
> > > We sincerely appreciate your recommendation of our paper! Your constructive comments and suggestions have helped us improve our paper a lot.

---

### Official Review · Reviewer_Kx4V · 2023-07-06

**Soundness:** 3 good
**Presentation:** 3 good
**Contribution:** 3 good
**Rating:** 6
**Confidence:** 5

**Summary:**

Inspired by the task of novel view synthesis, this paper additionally considers the audio modality and thus proposes an interesting new task: real-world audio-visual scene synthesis. Specifically, the task is to synthesize new videos with corresponding spatial audio along arbitrary novel camera trajectories by learning from a video recording of an audio-visual scene. To solve this problem, the authors resort to AV-NeRF, an A-NeRF for spatial audio synthesis, and a V-NeRF for video synthesis, respectively. For V-NeRF, they directly use the Vallina NeRF model. For A-NeRF, they explore extracting 3D geometry and material properties from the corresponding visual cues of V-NeRF. Moreover, a coordinate transformation module is proposed for more accurate spatial sound reconstruction. Better performance on a self-collected real-world dataset RWAVS and a simulation-based dataset SoundSpaces demonstrates the effectiveness of the proposed method AV-NeRF.

**Strengths:**

1. This paper proposes an interesting task: real-world audio-visual scene synthesis, which extends the task of novel view synthesis from only visual modality to multiple modalities.
2. The A-NeRF architecture introduced in this paper is well-designed. It considers sound propagation and view directions, which is quite reasonable. Besides, the coordinate transformation is also shown effective through comprehensive experiments.
3. Exhaustive experiments show the superior performance of AV-NeRF over several strong baseline methods.
4. The whole paper is well-written and presented in a coherent manner.

**Weaknesses:**

1. The design of the most important module AV-Mapper is not reasonable. First, the authors just use a pretrained ResNet-18 network to extract RGB and depth features from RGB and depth images. I'm curious how the ResNet-18 network could obtain material information from RGB images since the semantics can't imply the material properties at all. For a black desk, I can't tell if it's made of wood or steel just from the appearance. Actually, the authors can do some testing experiments, such as conducting color jittering and seeing what happens, or cross-scene testing to see whether the ResNet-18 network can extract materials and geometry information. Second, depth images can't inform all the geometry information. For example, the occlusion, which significantly influences sound propagation, can't be reflected on depth maps.
2. For spatial audio modeling, a useful metric that can really reflect the necessity of distance and view directions is required. Can the distance help to determine the energy of sounds on the receiver side? It's impossible to tell only from MAG and ENV metrics. The quantitative result w.r.t. **a new useful metric** should be provided in the paper. Similarly, I can't tell whether the view directions are useful for modeling the relative energy of left and right channels. I also wonder which one is more important for spatial audio modeling.

**Questions:**

1. How can the network obtain the distance between the camera and the sound source only from their positions? As we all know, distance calculation includes square and square roots, which are hard to approximate with simple neural networks.
2. The coordinate system in each scene. How to determine the origin?

**Limitations:**

1. The biggest limitation is the inappropriate modeling of AV-Mapper. The authors claim this module can obtain the material and geometry information from V-NeRF. However, from the perspective of this reviewer, such important information can't be inferred from the current design. Please refer to the weaknesses for details.
2. The current evaluation protocols can't validate the effectiveness of some modules employed in the paper.

---

> ### Author Rebuttal · Authors · 2023-08-09
>
> We thank the reviewer for your thoughtful review and valuable feedback. We address specific questions below.
>
> ### Weakness
> **W1: Rationality of AV-Mapper.** Great question! We leverage RGB and depth images as **implicit** indicators of environmental material properties and geometry. While these images enable the model to perceive the environment implicitly, it is important to note that achieving a precise one-to-one mapping from images to material properties or geometry is not guaranteed for AV-Mapper. Certain corner cases, such as a black desk and object occlusion as mentioned by the reviewer, can lead to the failure of AV-Mapper.
>
> Nevertheless, existing studies have demonstrated the feasibility of inferring material and geometry information from images. For instance, back in 2002, Varma and Zisserman [1] proposed a filter-based approach for material classification in images. More recently, in 2017, Zhang et al. [2] successfully recognized material and shape attributes using visual data. Moreover, there is a body of audio-visual research indicating that extracted geometry and material information via ResNet networks can contribute to audio synthesis [3, 4]. Our paper further supports this concept. As illustrated in Table 2 (left) of our paper, the inclusion of AV-Mapper in AV-NeRF can improve the MAG score by 16\% from 1.791 to 1.504 and enhance the ENV score by 3\% from 0.150 to 0.145.
>
> **W2: Necessity of distance and direction.** Thanks for the suggestion! To further validate the necessity of distance and direction, we conducted additional ablation studies. We intentionally exclude positional and directional coordinates when representing an auditory scene and evaluate the performance without these inputs. The results are presented below. AV-NeRF achieves a MAG metric score of 1.504 and an ENV metric score of 0.145 when both position and direction information are utilized. Removing either positional or directional input leads to a performance drop between 2\% to 21\%. In the absence of both positional and directional inputs, performance degrades further to 1.822 on the MAG metric and 0.156 on the ENV metric. These outcomes distinctly affirm the indispensable role of distance and direction coordinates in acoustic modeling.
>
> For metrics, we follow existing audio-visual learning work [5, 6, 7], utilizing MAG and ENV metrics to evaluate the quality of generated binaural audio. MAG and ENV metrics reflect the generation quality in time-frequency and time domain, respectively. We appreciate that the reviewer suggested developing a novel metric to quantify the influence of distance and direction on acoustic modeling and to establish their significance. It would be a very interesting idea to explore in the future.
>
> |Methods (Position) |Methods (Direction) |Overall (MAG) | Overall (ENV) |
> |:-:|:-:|:-:|:-:|
> | | | 1.822 | 0.156 |
> | | &check;| 1.817 | 0.155 |
> | &check;| | 1.688 | 0.148 |
> | &check;| &check;| **1.504** | **0.145**|
>
> ### Question
> **Q1: Distance between the camera and the sound source.** Audio-NeRF parameterizes the acoustic scene by learning a mapping from 5D coordinates $(x,y,z,\theta,\phi)$ to corresponding acoustic masks $\mathbf{m}_m, \mathbf{m}_d$ (Equation 5). The prediction of these acoustic masks solely relies on the query pose $(x,y,z,\theta,\phi)$. Audio-NeRF is designed to capture and learn acoustic effects caused by the distance, such as energy attenuation. However, it is not obligated to directly model and approximate the specific distance between the camera and the sound source. If we have misunderstood the reviewer's question, we kindly request further clarification, and we are more than willing to address any questions you might have.
>
> **Q2: Origin of the coordinate system.** We describe below the process of establishing the coordinate system for our camera poses $P=\{p_1, p_2, \dots, p_N\}$, where each pose is denoted as $p_i = (x_i, y_i, z_i, \theta_i, \phi_i)$, and $N$ represents the total number of camera poses. To ensure uniformity, we normalize the 3D coordinates $(x_i, y_i, z_i)$ to be within the range of $[-1, 1]\times[-1, 1]\times[-1, 1]$. This normalization is achieved by dividing each coordinate by the maximum value of all corresponding coordinates along the same axis. For instance, we normalize $x_i$ as $x_i^* = \frac{x_i}{\max_{k=1}^{N}|x_k|}$. This coordinate normalization practice is commonly used in training NeRF. Once all camera poses are normalized, we designate $(0, 0, 0)$ as the origin of our coordinate system.
>
> ### Limitation
> **L1: Please refer to W1.**
>
> **L2: Please refer to W2.**
>
> [1] Varma, Manik, and Zisserman, Andrew. "Classifying images of materials: Achieving viewpoint and illumination independence." ECCV. 2002.
>
> [2] Zhang, Zhoutong, et al. "Generative modeling of audible shapes for object perception." ICCV. 2017.
>
> [3] Chen, Changan, et al. "Learning audio-visual dereverberation." ICASSP. 2023.
>
> [4] Singh, Nikhil, et al. "Image2reverb: Cross-modal reverb impulse response synthesis." ICCV. 2021.
>
> [5] Gao, Ruohan, and Grauman, Kristen. "2.5 d visual sound." CVPR. 2019.
>
> [6] Xu, Xudong, et al. "Visually informed binaural audio generation without binaural audios." CVPR. 2021.
>
> [7] Chen, Changan, et al. "Novel-view acoustic synthesis." CVPR. 2023.

---

> > ### Comment · Reviewer_Kx4V · 2023-08-18
> > **I have read authors' rebuttal**
> >
> > Thanks so much for your detailed response, which has already addressed most of my concerns. I'm happy to accept this paper and therefore decide to raise my score. Please include these additional results and discuss the limitations of AV-Mapper in the revision. By the way, for the distance between the camera and the sound source, I mean the root or square root operation may be hard to approximate with simple neural networks (just two layers for example), and wonder about the detailed neural network architectures.

---

> > > ### Author Response · Authors · 2023-08-18
> > >
> > > We thank the reviewer for recommending the acceptance of our paper! We will revise the paper to include additional results and discuss AV-Mapper’s limitations.
> > >
> > > We agree with the reviewer that the root or square root is difficult to approximate with simple networks. However, AV-NeRF does not need to explicitly model the distance between the camera and the sound source. Therefore, this issue does not apply to AV-NeRF.
> > >
> > > For model architectures, we provide a detailed description below:
> > > 1. A-NeRF consists of two Multilayer Perceptrons (MLPs), each with four linear layers and an additional residual connection. The width of each linear layer is set to 128 for the RWAVS dataset and 256 for the SoundSpaces dataset.
> > > 2. In V-NeRF, we design a 64-width 2-layer MLP for density modeling and a 64-width 3-layer MLP for color modeling.
> > > 3. The AV-Mapper is implemented as a 3-layer MLP, with hidden dimensions decreasing gradually from 512 to 128.
> > > 4. For all MLPs, we use ReLU as the activation function.
> > >
> > > We have provided the detailed information in the appendix, but we are happy to move it to the main paper if that is preferred. For reproducibility, we will release our source code, data, and models.

---

### Official Review · Reviewer_GFnA · 2023-07-07

**Soundness:** 3 good
**Presentation:** 2 fair
**Contribution:** 2 fair
**Rating:** 5
**Confidence:** 4

**Summary:**

This paper describes an intriguing task that focuses on the synthesis of new perspectives of real-world audiovisual scenes. The task is to synthesize a new video with spatial audio along an arbitrary novel camera track in an audiovisual scene given a video recording of that audiovisual scene. A data acquisition system is built and a Real-World Audio-Visual Scene (RWAVS) dataset is collected in this paper.
First, the paper proposes an acoustically aware audio generation module that integrates prior knowledge of audio propagation into NeRF, thus relating audio generation to the 3D geometry of the visual environment. In addition, a coordinate transformation module that represents the direction of observation with respect to the sound source is proposed. This directional transformation aids the model in learning the source-centered sound field. This paper's superiority is demonstrated by qualitative and quantitative results.

**Strengths:**

1. The synthesis of new views on audiovisual scenes in the real world is an interesting task that contributes to a better understanding of the world and the dataset presented can contribute to the development of the field.

2. The paper is well written and each module is clearly presented and easy to understand.

3. The proposed AV-Mapper and the coordinate transformation mechanism can effectively fuse visual geometric information with audio information and effectively express sound direction.


**Weaknesses:**

1. The vanilla NeRF [15] structure is used in this paper's V-Nerf module, which is a relatively slow training and sampling model. Existing acceleration models for Nerf can already significantly improve training and sampling speed, and a better model may be able to achieve higher accuracy in this paper.

2. How does the proposed model handle sounds that are not within the field of view?

3. There are numerous audio-visual datasets available, such as the AVSBench (https://github.com/OpenNLPLab/AVSBench). What is the proposed method's performance on this dataset?



**Questions:**

As above.

**Limitations:**

As above.

---

> ### Author Rebuttal · Authors · 2023-08-09
>
> We thank the reviewer for your helpful suggestions and encouraging comments. We address specific comments below.
>
> ### Weakness
> **W1: NeRF acceleration.** We concur with the reviewer's observation that one limitation of the vanilla NeRF is its slow rendering speed. In our study, we actually employ acceleration techniques for NeRF to enhance both the training and rendering processes. Sorry for the misunderstanding! We would like to further clarify the implementation details here. Specifically, we incorporate the `tiny-cuda-nn` library, which facilitates rapid NeRF training and querying [1, 2]. Furthermore, we utilize `nerf-studio` [3] to enhance rendering quality, as detailed in L23-L28 of the supplementary material. `nerf-studio` offers support for camera pose refinement, image appearance conditioning, hash encoding, and proposal sampling.
>
> **W2: Out-of-view sound.** Audio-NeRF parameterizes the acoustic scene by learning a mapping from 5D coordinates $(x,y,z,\theta,\phi)$ to corresponding acoustic masks $\mathbf{m}_m$ and $\mathbf{m}_d$ (see Equation 5). The prediction of acoustic masks is solely based on the query pose $(x,y,z,\theta,\phi)$. The RGB and depth images input into AV-Mapper serve as implicit indicators of the environment's material properties and geometry. Therefore, the absence of sound sources in the field of view has no impact on mask prediction.
>
> **W3: Results on AVSBench dataset.** We appreciate the reviewer for proposing the AVSBench dataset as a valuable resource. We acknowledge the significance of this dataset within the audio-visual community and will cite it in our revised manuscript. However, we would like to note that the AVSBench dataset may not align with our specific data requirements. Our proposed AV-NeRF model necessitates access to ground-truth audio data from both the sound source and the sound receiver. Regrettably, the videos provided by the AVSBench dataset lack the essential ground-truth sound information pertaining to the sound source. Furthermore, the videos encompass dynamic scenes, posing challenges for accurate camera pose estimation. Due to these limitations, conducting experiments using the AVSBench dataset becomes impractical for our study.
>
> [1] Müller, Thomas. Tiny CUDA Neural Network Framework.
>
> [2] Müller, Thomas, et al. "Instant neural graphics primitives with a multiresolution hash encoding." ACM Transactions on Graphics (ToG) 41.4 (2022): 1-15.
>
> [3] Tancik, Matthew, et al. "Nerfstudio: A modular framework for neural radiance field development." ACM SIGGRAPH 2023 Conference Proceedings. 2023.

---

> > ### Comment · Reviewer_GFnA · 2023-08-21
> >
> > My concerns have been addressed.
> >
> > My only concern is the reproducibility of the proposed method. Therefore, I keep my initial rating.

---

> > > ### Author Response · Authors · 2023-08-21
> > >
> > > We thank the reviewer for your helpful comments. For reproducibility, we will release our source code, data, and models.

---

### Official Review · Reviewer_b4XE · 2023-07-07

**Soundness:** 3 good
**Presentation:** 3 good
**Contribution:** 3 good
**Rating:** 5
**Confidence:** 4

**Summary:**

The paper proposes an audio-visual (AV-) NeRF model to synthesize binaural audio masks at novel poses by using audio-visual samples from a video walkthrough of a 3D scene. The synthesized audio masks can be convolved with any arbitrary anechoid audio signal to retrieve the corresponding binaural audio at the novel poses. Towards that goal, the paper proposes a model that uses the learned density information from a visual (V-) NeRF model to provide geometric and color information at a novel pose. Given this visual information, an audio (A-) NeRF model predicts masks that can be further used to generate the synthesize the binaural audio for a monaural sound source. Additionally, the paper proposes an alternate parameterization to the direction component of the camera pose parameters that accounts for the omnidirectionality of most sound sources. The paper also collects a realworld dataset for the task. The paper evaluates on both simulated and realworld data, compares against multiple baselines and reports impressive results.


Post author-reviewer discussion: I have read the rebuttal. It addresses my concerns. I will keep my score and recommend accepting the paper.

**Strengths:**

1, The paper proposes a new AV-NeRF model for binaural audio synthesis that leverages learned color and geometry information for novel poses from a V-NeRF model.

2. The alternate parameterization for the camera pose parameters is also a useful contribution that captures the physics of how sound travels from a omnidirectional sound source.

3. The paper also introduces a new realworld dataset for the task, which could be a valuable contribution for the community.

4. The paper evaluates on both simulated and realworld data, compares against multiple baselines and reports impressive results.

**Weaknesses:**

1. The paper doesn't compare with recently published ViGAS [14] model. The paper argues that ViGAS needs GT images and is limited to a few viewpoints. However, the images rendered by a standalone V-Nerf can be used as input to ViGAS (since it's not a contribution of this paper anyway), and ViGAS can be trivially adapted to render spatial sounds at arbitrary points.

Minor:
1. L131: [40] generalizes to new scenes, unlike the model proposed in this paper (also pointed out by the authors in limitations). Since there are no experiments on novel scenes, this statement looks unsubstantiated.

**Questions:**

I would like to request the authors to address the concerns I mentioned in 'Weaknesses'.

**Limitations:**

The paper already discusses its limitations and societal impact.

---

> ### Author Rebuttal · Authors · 2023-08-09
>
> We thank the reviewer for your constructive comments and encouraging remarks. We address specific comments below.
>
> ### Weakness
> **W1: Compare with ViGAS.** Because ViGAS is a concurrent work to our paper, we did not compare AV-NeRF with ViGAS in our submitted paper. However, in response to the reviewer's suggestion, we conduct an evaluation of ViGAS on the RWAVS dataset for performance comparison. We utilize the official open-source code available on GitHub and make minimal adjustments to the framework, enabling support for arbitrary camera poses. The obtained results of ViGAS's performance in each scene, as well as the overall performance, are presented below.
>
> | Methods | Office (MAG) | Office (ENV) | House (MAG) | House (ENV) | Apartment (MAG)|Apartment (ENV)|Outdoors (MAG)|Outdoors (ENV) |Overall (MAG)|Overall (ENV)|
> |:-|:-:|:-:|:-:|:-:|:-:|:-:|:-:|:-:|:-:|:-:|
> |INRAS | 1.405 | 0.141 |  3.511 |  0.182 |  3.421 | 0.201 |  1.502 |  0.130 |  2.460 |  0.164 |
> |NAF | 1.244 | 0.137 | 3.259 | 0.178 | 3.345 | 0.193 | 1.284 | 0.121 | 2.283 | 0.157 |
> |ViGAS | 1.049 | 0.132 | 2.502 | 0.161 | 2.600 | 0.187 | 1.169 | 0.121 | 1.830 | 0.150 |
> |Ours | **0.930** | **0.129** | **2.009** | **0.155** | **2.230** | **0.184** | **0.845** | **0.111** | **1.504** | **0.145** |
>
> We can see that ViGAS achieves pretty good performance, and it outperforms both INRAS and NAF, serving as a strong baseline.
> Our approach is better than ViGAS in terms of both MAG (1.504 compared to ViGAS's 1.830) and ENV (0.145 compared to ViGAS's 0.150). These results further underscore the effectiveness of our method in accurately capturing the underlying acoustic field.
>
> **W2: Task definition.** We agree with the reviewer's point regarding the statement in our task definition. [1] can generalize to new scenes. In our revision, we will modify the task statement to address this concern.
>
> [1] Majumder, Sagnik, et al. "Few-shot audio-visual learning of environment acoustics." NeurIPS. 2022.

---

> > ### Comment · Reviewer_b4XE · 2023-08-17
> >
> > The rebuttal addresses my concerns. I will keep my score and recommend accepting the paper.

---

> > > ### Author Response · Authors · 2023-08-17
> > >
> > > We thank the reviewer for recommending the acceptance of our paper. We sincerely appreciate your constructive comments. We will include ViGAS's results in our revised paper.

---

### Author Rebuttal · Authors · 2023-08-09

We would like to express our gratitude to all the reviewers for their valuable comments and feedback. We address the specific questions of each reviewer individually. Additionally, we attach a PDF file containing some figures in response to certain reviewers.

**To reviewer 4psZ:** we show some additional results in Figure 2 of the attached one-page PDF file.

**To reviewer iWmP:** we show failure cases in Figure 1 of the attached one-page PDF file. Due to the word limit of rebuttal, we place the reference paper here:

[1] Richard, Alexander, Peter Dodds, and Vamsi Krishna Ithapu. "Deep impulse responses: Estimating and parameterizing filters with deep networks." ICASSP. 2022.

[2] Luo, Andrew, et al. "Learning neural acoustic fields." NeurIPS. 2022.

[3] Gao, Ruohan, et al. "Visualechoes: Spatial image representation learning through echolocation." ECCV. 2020.

---

### Decision · Program_Chairs · 2023-09-21

**Decision:**

Accept (poster)

**Comment:**

All reviewers recommend acceptance. The AC sees no basis to overturn the reviews. Authors should attend to main points in the reviews, such as the concerns about evaluation (e.g. Reviewer iWmP), when preparing a final version.